**Data Availability Statement:** The dataset that was used is held by the National Health Insurance Fund (NHIF) of Hungary (http://www.neak.gov.hu, e-

# Epidemiology, mortality and prevalence of colorectal cancer in ulcerative colitis patients between 2010-2016 in Hungary – a population-based study

**Péter Kunovszki**[1]☯*, **Ágnes Milassin**[2]☯, **Judit Gimesi-Országh**[1], **Péter Takács**[1], **Kata Szántó**[2], **Anita Bálint**[2], **Klaudia Farkas**[2], **András Borsi**[3], **Péter L. Lakatos**[4,6], **Tamás Szamosi**[5], **Tamás Molnár**[2]

**1** Janssen Global Commercial Strategy Organization, Budapest, Hungary, **2** First Department of Internal Medicine, University of Szeged, Szeged, Hungary, **3** Janssen-Cilag Limited, High Wycombe, England, United Kingdom, **4** Semmelweis University, Budapest, Hungary, **5** Military Hospital-State Health Centre, Budapest, Hungary, **6** Division of Gastroenterology, McGill University, Montreal, Canada

☯ These authors contributed equally to this work.
* pkunovsz@its.jnj.com

## Abstract

### Background

The incidence and prevalence of ulcerative colitis (UC) varies geographically. The risk of colorectal cancer (CRC) and possibly some other malignancies is increased among patients with UC. It is still debated if patients with UC are at a greater risk of dying compared with the general population. Our aim was to describe the epidemiology and mortality of the Hungarian UC population from 2010 to 2016 and to analyze the associated malignancies with a special focus on CRC.

### Methods

This is an observational, descriptive, epidemiological study based on the National Health Insurance Fund social security databases from 2010 to 2016. All adult patients who had at least two events in outpatient care or at least two medication prescriptions, or at least one inpatient event with UC diagnosis were analyzed. Malignancies and CRC were defined using ICD-10 codes. We also evaluated the survival of patients suffering from UC compared with the general population using a 3 to 1 matched random sample (age, gender, geography) from the full population of Hungary.

### Results

We found the annual prevalence of UC 0.24–0.34%. The incidence in 2015 was 21.7/100 000 inhabitants. Annual mortality rate was 0.019–0.023%. In this subpopulation, CRC was the most common cancer, followed by non-melanotic skin and prostate cancer. 8.5% of the UC incident subpopulation was diagnosed with CRC. 470 (33%) of the CRC patients died during the course of the study (25% of all deaths were due to CRC), the median survival was

mail: neak@neak.gov.hu). Access to the individual-level data is available after filing a formal data access request to adatkeres@neak.gov.hu. Requestors need to accept the terms and conditions of the data request and may need to pay the corresponding data access fee. The terms of the contract for data access does not allow the reporting of any data of a single individual or results which comes from aggregating the data of less than 10 individuals. Therefore, a de-identified dataset could not be provided. Taking these requirements into consideration, the results can be published. A supplementary dataset was created which contains the patient counts derived from the original data.

**Funding:** The study was funded by Janssen: Pharmaceutical Companies of Johnson & Johnson (www.janssen.com). PK, JGO, PT, AB are employees / consultants of Janssen. The funders had no role in study design, data collection and analysis, decision to publish, or preparation of the manuscript.

**Competing interests:** I have read the journal's policy and the authors of this manuscript have the following competing interests: PK, JGO, PT, ABo are employees / consultants of Janssen. PLL has been a speaker and/or advisory board member for AbbVie, Arena Pharmaceuticals, Celltrion, Falk Pharma GmbH, Ferring, Genetech, Janssen, Merck, Pharmacosmos, Pfizer, Roche, Shire and Takeda and has received unrestricted research grants from AbbVie, MSD and Pfizer. TSz has served as advisory board member for AbbVie, EGIS, Pfizer and Takeda, received speaker's honoraria from Abbvie, Takeda and Ferring and served as part time medical advisor for Hungarian National Health Insurance Fund. TM received speaker's honoraria from MSD, AbbVie, Egis, Goodwill Pharma, Takeda, Pfizer and Teva. KF received speaker's honoraria from AbbVie, Janssen and Ferring. ABá: received speaker's honoraria from Janssen and Ferring. KSz, ÁM have no conflicts of interest to declare. This does not alter our adherence to PLOS ONE policies on sharing data and materials.

9.6 years. UC patients had significantly worse survival than their matched controls (HR = 1.65, 95% CI: 1.56–1.75).

## Summary

This is the first population-based study from Eastern Europe to estimate the different malignancies and mortality data amongst Hungarian ulcerative colitis patients. Our results revealed a significantly worse survival of patients suffering from UC compared to the general population.

## Introduction

Ulcerative colitis (UC) is a chronic inflammation of the colon with an unknown etiology. Inflammatory bowel disease (IBD) is associated with morbidity, mortality, and substantial costs to healthcare systems, therefore several studies have attempted to define the burden of the disease. The occurrence of ulcerative colitis is changing in the 21st century. The incidence and prevalence rates vary geographically. Previous studies report the highest incidence rates from industrialized countries, such as North America and Western Europe [1, 2]. The population-based study of the ECCO EpiCom-group (European Crohn's and Colitis Organization–Epidemiological Committee) revealed an east-west gradient in the incidence of IBD. They found higher incidence rates in Western European countries, and lower ones in Eastern European countries (except for Hungary), where the incidence rates were similar to the Nordic countries. [3, 4]. Almost all countries had higher incidence rates in 2010 in contrast to the older studies. These findings were validated by the 2011 ECCO-EpiCom inception cohort study with similar results. [3, 4]. The incidence of UC has increased over the second part of the 20th century in many areas with formerly low incidence rates, but it has stabilized in high-incidence areas [1, 5, 6]. According to previously published Hungarian studies, the incidence of UC increased from 1977 to 2001, and then stabilized, while the prevalence rate increased from 2001 to 2006 [7, 8]. The last prevalence data (0.34% for UC) originates from a study between 2011 and 2013 based on the National Health Insurance Fund (NHIF) database [9].

There is a gap in knowledge about whether patients suffering from UC have a higher mortality risk compared to the general population. A meta-analysis found the overall mortality similar to the general population [10].

The aim of our study was to describe the incidence, prevalence rate, and the mortality of the Hungarian UC population from 2010 to 2016, and to analyze the prevalence of malignancies with a special focus on colorectal cancer (CRC). The mortality data of the UC population was compared to that of the general population to determine mortality risk.

## Materials and methods

### Data collection

This is an observational, non-interventional, retrospective, descriptive, epidemiological study based on the National Health Insurance Fund (NHIF) social security database. This database contains financial claims data on all healthcare events of the whole population of Hungary, a population of approximately 10 million people. These include inpatient hospital stays, outpatient visits, pharmacy drug reimbursements and special drug reimbursements. Unlike similar databases from other countries, medication prescriptions carry diagnosis information as well.

It also includes demographic data on the population (date of birth, gender, geographical region and date of death, where applicable). On the other hand, non-finance-related information–such as laboratory test results–are not available. The data was analyzed between 2010 and 2016.

Differentiation between Crohn's disease (CD) and UC using a financial claims database is a complex task, because a considerable number of patients can exist in the database who have both diagnosis codes at the same time. This can happen because of the difficult differentiation during the early stages of disease.

As a first step, a background population consisting of patients with either CD or UC was created. Those patients were selected who had at least two events among all relevant health care services, or at least one inpatient event with the diagnosis of UC (based on the 10th revision of the International Statistical Classification of Diseases and Related Health Problems (ICD-10) code: K51*) or CD (ICD-10 code: K50*) during the study period between 2010 and 2016. Back-data was available for patients from the start of 2007. The next step was to determine which of those patients with both diagnoses could be ascertained as UC patients. These patients along with those who had only UC diagnoses formed the study population. The algorithm was as follows: a patient had to have at least 80% majority of UC diagnosis codes to be classified as an UC patient.

The patient group still contained patients for whom an UC diagnosis could not be ascertained, so a further refining step, similar to the one outlined in the paper by Kurti et al, was necessary [9]. In this step patients were excluded if they had none of the following: any record of biological therapy (BT), any record of UC-related surgical interventions (based on Diagnosis-related group (DRG)), or record of sufficient number of drug prescriptions (defined as at least 2 dispensings of 5-aminosalicylates (5-ASA) or corticosteroids (CS) or immunosuppressants (IS) per year). This requirement on the number of prescriptions causes that patients with a short follow-up time–i.e. those who are incident in the second half of 2016 –to be easily excluded from the analysis. Therefore, the year of 2016 was excluded from the epidemiology analysis.

Survival analysis was performed on a subgroup of patients who were newly diagnosed (defined as having no UC diagnoses before) from the beginning of 2010. This means that all incident patients have at least a 3-year long UC diagnosis-free baseline period.

For comparison purposes, date of death data of a 3 to 1 matched reference population from the total Hungarian population was obtained. The matching was performed based on age (year of birth), gender and permanent residency. No information other than the date of death could be obtained for these people, therefore no other analyses regarding these controls were possible.

To analyze the incidence, the prevalence and the mortality data of the UC population, point prevalence was used at the first day of each year. Demographic data was also evaluated.

Malignant neoplasms were evaluated in the incident UC subpopulation. Malignancies were categorized based on 3-digit ICD10 codes, presence of a certain malignancy was defined as having at least 2 diagnoses in the in- or outpatient care setting after the UC diagnosis date. The presence of colorectal cancer (CRC) was analyzed in detail. CRC was defined using ICD-10 codes C15*-C26*. The first CRC code appearance was considered as the date of the diagnosis of CRC. Proportion of patients with certain malignancies, yearly incidence and prevalence of CRC was evaluated. Survival data of the study population were assessed. Overall survival from the time of the diagnosis of UC and from the time of the diagnosis of CRC were evaluated. Overall survival data from the diagnosis of UC were compared with the data from the matched general population.

## Statistical analysis

Prevalence, incidence and mortality was described using patient counts. Demographic data was characterized using histograms and median age. To compare the age of different patient groups t-tests were used.

Survival analysis was performed to analyze overall survival, Kaplan-Meier estimators were used to characterize the survival function.

Comparison of survival of two groups is performed in two cases, for all UC patients versus controls and for the two different age groups for the CRC subpopulation. The survival curves were compared using log-rank tests. A Cox proportional hazards model–using a single binary predictor for the two groups–was also used for comparison. The hazard ratio with 95% confidence interval between the two groups was given as a result. The proportional hazards assumption was tested using plots of Schoenfeld residuals.

Due to the claims nature of the data, missing data is undiscoverable in most cases. If a certain intervention or diagnosis was not recorded, there is no chance that it could be imputed in any way. Therefore, no handling of missing data was performed.

Analyses was carried out using the statistical software R 3.5.1 (R Core Team (2018). R: A language and environment for statistical computing. R Foundation for Statistical Computing, Vienna, Austria. URL https://www.R-project.org/).

## Ethical approval

This study has been approved by Medical Research Council–Research and Ethics Committee (TUKEB), Hungary (Appr. no: 12288-3/2018/EKU).

All data used in the study was held by NHIF, the researchers had access only to anonymized data. Data protection guidelines did not permit the reporting of patient level data even in anonymized form, only aggregate results could be reported.

The database contains information on the full population of Hungary (approximately 10 million people), the total number of patients whose data entered the patient selection algorithm was 224 175. The study population consisted of 37 795 patients after applying all inclusion and exclusion criteria.

## Results

### Epidemiology of UC, demographics

For the reasons outlined in the Materials and methods section, the incident patients in 2016 were excluded from the epidemiology analysis.

The number of patients suffering from UC between 2010–2015 was 36 315. The annual prevalence increased during the examined period (Fig 1). In 2010 0.24%, while in 2015 0.34% of the total Hungarian population suffered from UC.

A decreasing trend was found in the incidence of UC within the study period. This could be at least partly attributed to the way the diagnosis was defined, which is a limitation of our study. In 2010 there was only a 3-year baseline period for patients, while in 2015 this baseline period was longer (8 years). This causes that patients incident in real life before 2007 with longer gaps in their medical history of UC have a chance to be identified as an incident patient at the beginning of the study but this probability is much smaller in the later years. Therefore, the least biased estimate of incidence is the one from 2015 –as 2016 had to be excluded for reasons outlined in the Materials and methods section. This estimated incidence of UC from 2015 was 21.7/100 000 inhabitants.

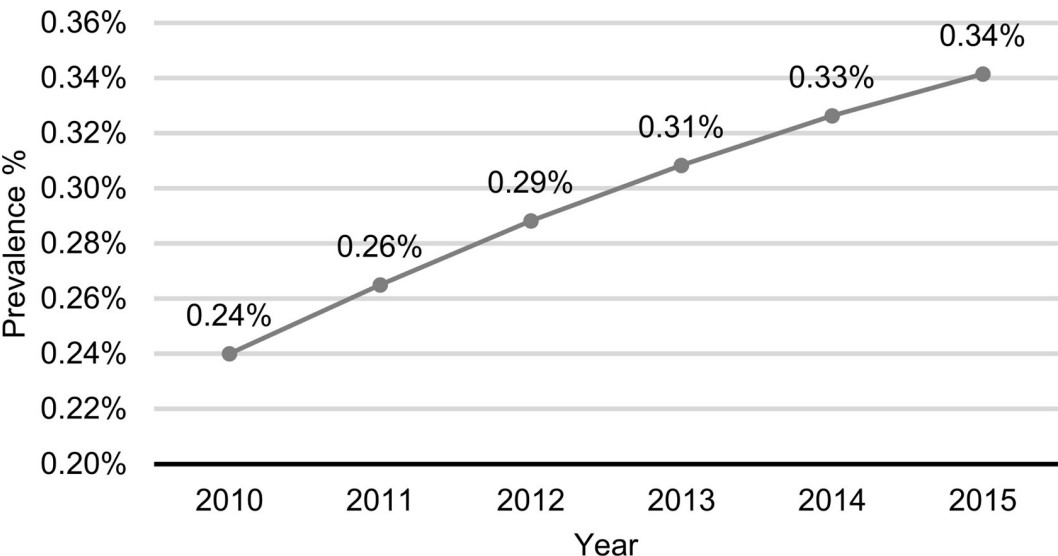

**Fig 1. The annual prevalence of ulcerative colitis in Hungary.**

The proportion of females amongst the prevalent population was 55%. The median age of patients at the time of the first diagnosis of UC was 51 years (males 49, females 53). The average and median age was higher for women which is demonstrated clearly on the population pyramid (Fig 2). The difference is statistically significant at p<0.001.

In total 3 188 prevalent patients died in the study period between 2010 and 2015. The annual mortality rate between 2011 and 2015 was stable, varying between 18.7 and 23.3 per 1000 patients. The rate was considerably lower in 2010 (12.8/1000 patients), though (Fig 3).

The median age at the time of death was 75.3 years in the whole UC population. Men died at a younger age (median 71.9) than women (median 78.3) which is in concordance with the trend observable in the general population of Hungary.

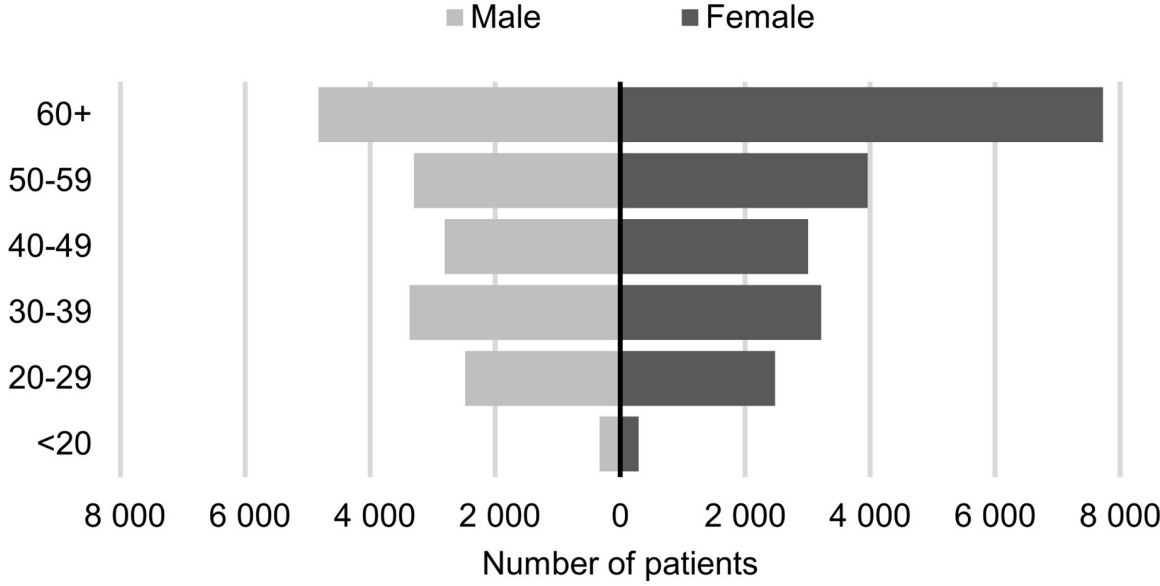

**Fig 2. The population pyramid (distribution by age and gender) of the prevalent UC population at the time of diagnosis.**

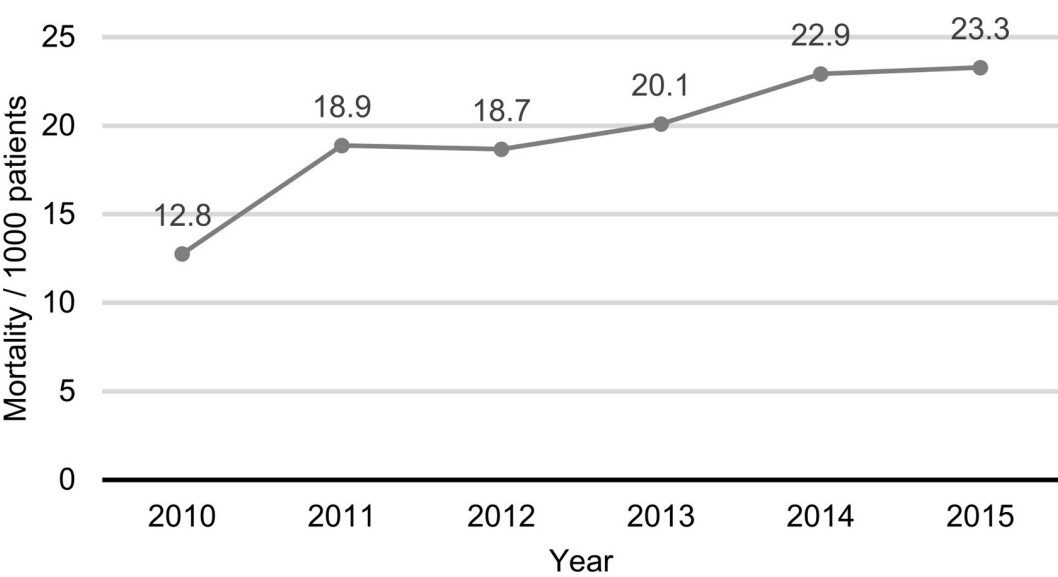

**Fig 3. Mortality rate of UC patients between 2010 and 2015.**

### Malignancies of UC patients

There were 16 712 patients who were diagnosed for the first time during the observational period (incident population). Investigating all malignant neoplasms of this incident UC population, CRC was found to be the most common cancer followed by non-melanotic skin cancer and prostate cancer (Fig 4).

As CRC was the most commonly appearing malignancy, and because it can be the effect of long-term UC, it was analyzed in detail. In total 1 424 patients (8.5%) were diagnosed with CRC in the incident patient subpopulation. The number of new diagnoses was stable within the study period with roughly 200 new cases every year (Fig 5). A small decrease in the numbers is observable in 2015 and 2016. These results may be biased due to the methodology where two diagnoses of CRC were required for a patient to be considered. With shorter follow-up times the probability of a second diagnosis appearing can be lower.

Among patients with CRC 470 (33%) have died, these deaths make up 25% of all deaths within the incident UC population.

The median age of patients at the time of CRC diagnosis was 65.8 years (male: 64.7; female: 67.0). The median age of these patients at the time of death was 71.1 years (male: 68.9; female: 73.3). These patients died at a younger age than the average patient with UC as the median age at the time of death within the incident UC population was 75.3 years (male: 71.9; female: 78.3).

### Survival of UC and CRC patients

Overall survival of the incident UC patients from the time of diagnosis was examined (Fig 6). The survival probability decreased with increasing time elapsed at a linear rate. The 1-year survival rate was 97%, the 3-year survival rate was 91% and the 5-year survival rate was 86%.

UC patients have significantly worse survival than their matched controls (HR = 1.65, 95% CI: 1.56–1.75).

Overall survival of CRC patients amongst the UC patients from the CRC diagnosis was also analyzed (Fig 7). The survival probability decreased with increasing time elapsed at a linear rate. The 1-year survival rate was 88%, the 3-year survival rate was 75% and the 5-year survival rate was 65%. The median survival was 9.67 years.

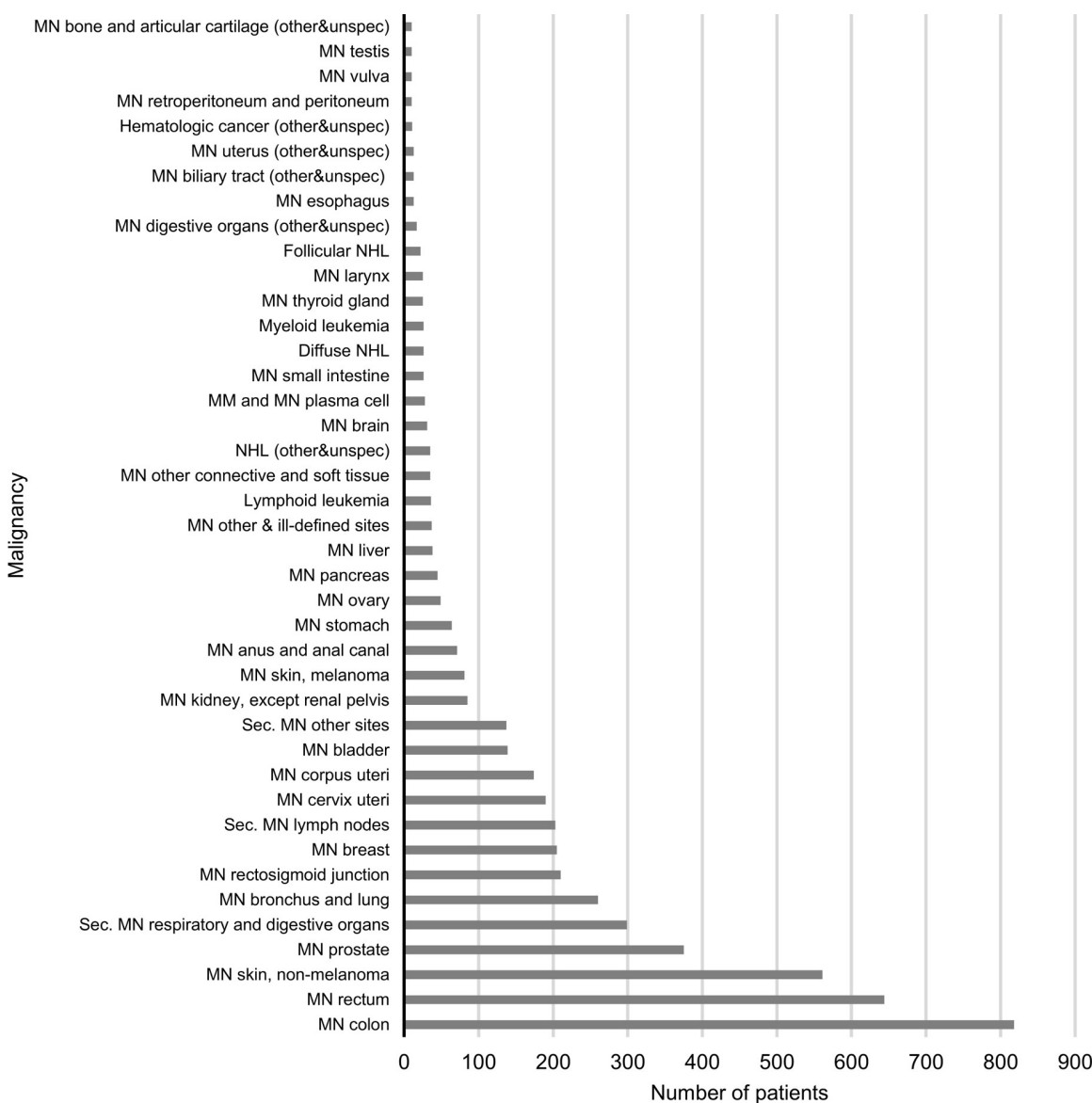

**Fig 4. The number of patients diagnosed with certain types of malignant cancers.** MN–malignant neoplasm of, Sec. MN–secondary malignant neoplasm of, (other&unspec)–other or unspecified part/type, NHL–non-Hodgkin's lymphoma, MM–multiple myeloma.

This analysis was also performed using a breakdown of patients based on age (over and under 60 years) (Fig 8). No significant difference could be found between the survival probabilities of these two age groups (HR = 1.18, 95% CI: 0.94–1.46, p = 0.147).

## Discussion

This is the first population-based study from Eastern Europe, which simultaneously estimates the prevalence and incidence rates, the mortality and the morbidity and the associated malignancy data based on the Hungarian National Health Insurance Fund database. In the present study, the prevalence of UC in the Hungarian population increased from 0.24% to 0.34%. Based on previous Hungarian studies, the prevalence rate of UC was 10.4/100 000 people from 1962–1992, 142.6/100 000 from 1991 to 2001 and 211.1/100 000 from 2002–2006 (that

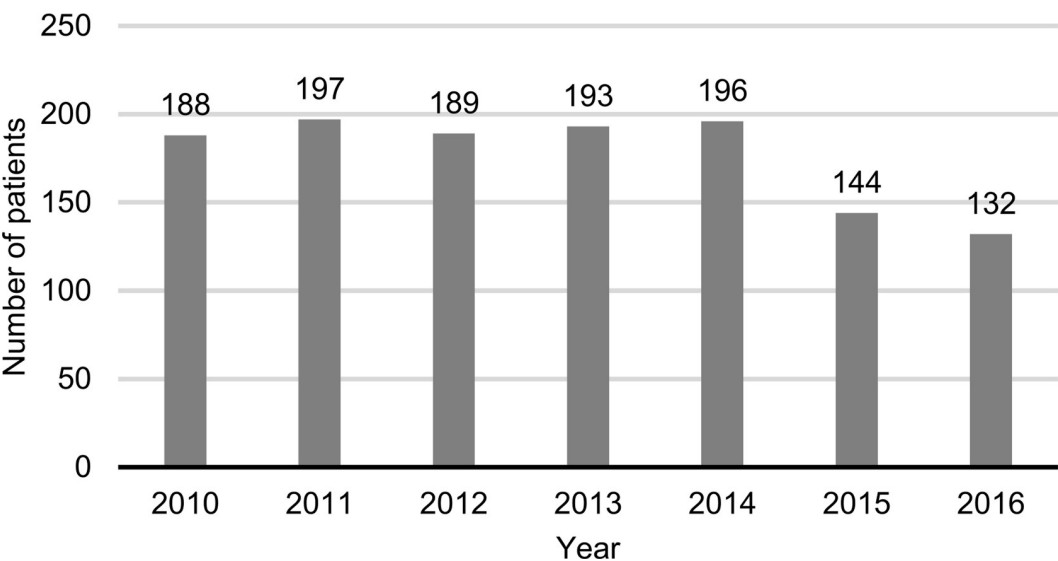

**Fig 5. Incidence of CRC in the incident UC population.** Diagnosis of UC and CRC is not necessarily in the same year.

corresponds to 0.01% between 1962 and 1992, to 0.14% between 1991 and 2001, and to 0.21% between 2002 and 2006) [7, 8, 11]. A Hungarian population-based study between 2011 and 2013 based on the National Health Insurance Fund database found a prevalence of 0.34% for

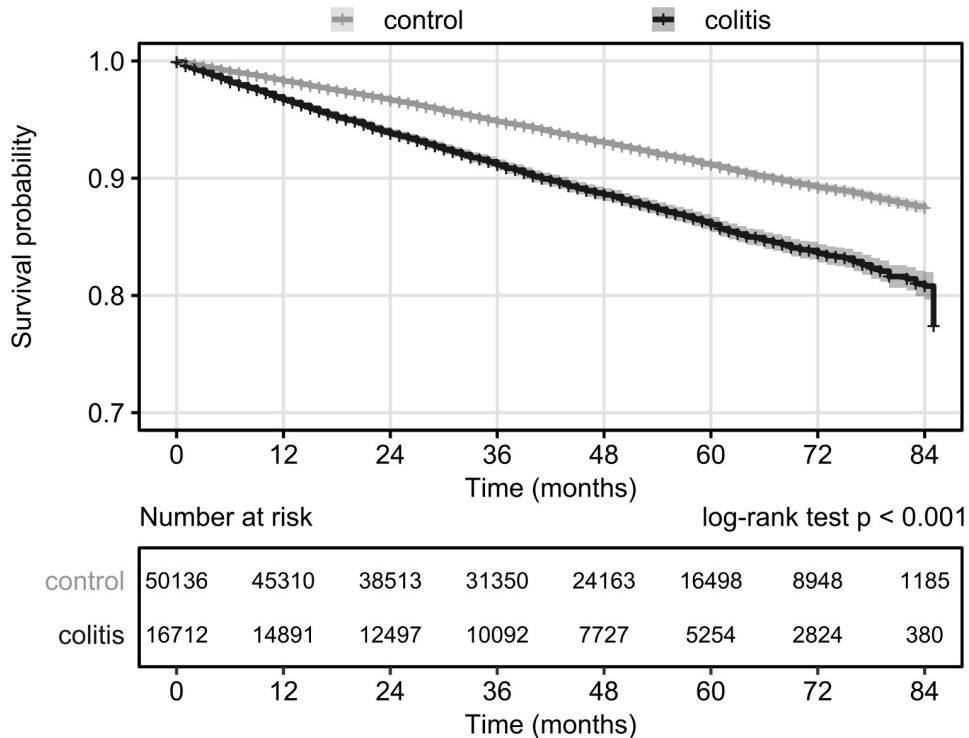

**Fig 6. Overall survival of UC patients and matched controls from diagnosis.** Start: UC diagnosis date for UC patients, UC diagnosis date of matched patients for controls. Event: death. Censoring: end of data availability (end of 2016). Shaded areas denote 95% confidence bands.

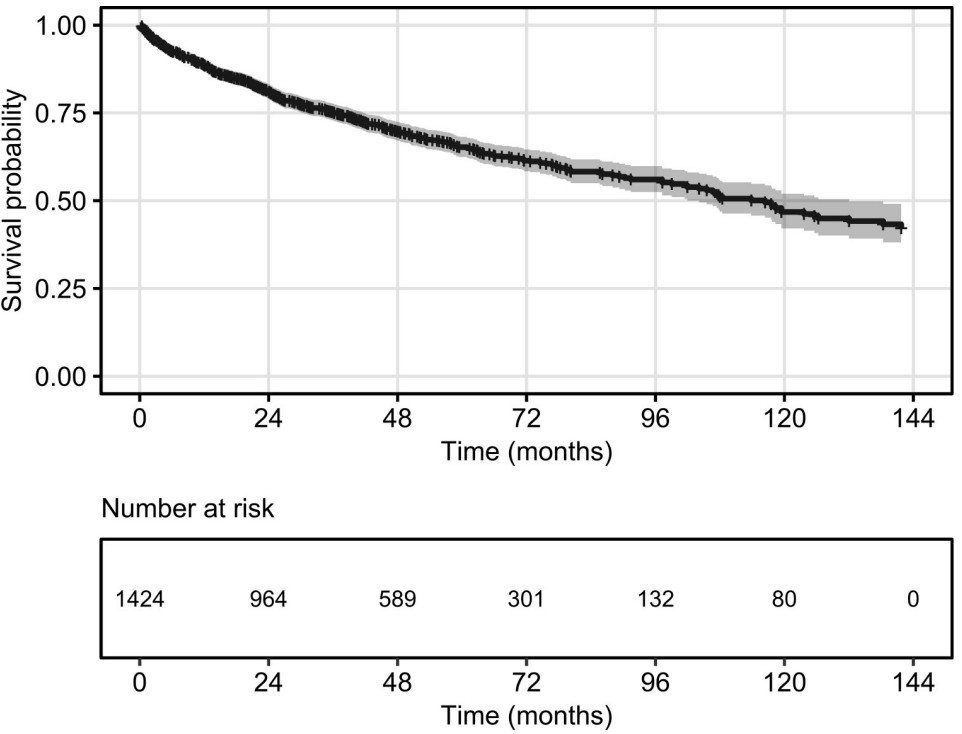

**Fig 7. Overall survival of CRC patients amongst the UC patients from CRC diagnosis.** Start: CRC diagnosis date. Event: death. Censoring: end of data availability (end of 2016). Shaded area denotes 95% confidence band.

UC patients [9], which is similar to our findings. They found the highest prevalence in Western Hungary (0.49%), and in the South-West region (0.35%).

Based on previous Hungarian studies, the incidence rate of UC has been increasing, it was 1.4/100 000 from 1962 to 1992, 5.89/100 000 from 1997 to 2001, 11.9/100 00 from 2002 to 2006 [7, 8, 11]. The incidence of 21.7/100 000 inhabitants in 2015 found in our study is considerably higher than the one reported in 2006. The increasing incidence tendency is in concordance with the published data from industrialized countries. The population-based study of the ECCO EpiCom-group revealed an east-west gradient, the median crude annual incidence rates for UC were 10.8 cases per 100 000 persons in 2010 in Western European centers, while in Eastern European centers (except Hungary) it was lower (4.1/100 000). Interestingly, in Hungary the incidence rates were similar to the high incidence in Nordic countries. In the 2011 ECCO-EpiCom inception cohort the mean annual incidence rates were lower, however, on the individual center level the results corresponded to the findings in the 2010 inception cohort [3, 4]. In a population-based study of French adolescents, UC incidences increased from 1988–1990 to 2009–2011 from 1.6 to 4.1/100 000 [12]. The multicenter European Collaborative Study on Inflammatory Bowel Disease (EC-IBD) reported blended incidence rates between 8.7–11.8 cases per 100 000 person-years for UC [2]. Lower incidence and prevalence rates can be observed in other Middle and Eastern European countries (Czech Republic: 5.5/100 000 in 2010, Poland: 1.8/100 000 between 1990 and 2003, Romania: 0.97/100 000 between 2002 and 2003, Slovakia: 6.8/100 000 in 2013) [3, 13, 14, 15].

Most studies reported the peak incidence of UC in the early adulthood period (in the second to fourth decade); however, in some studies a second modest rise in incidence in latter decades of life was reported (between 50–70 years old) [5]. In some previous studies published from Hungary the peak onset age was between 21 and 40 years [8]. In contrast, two peaks in

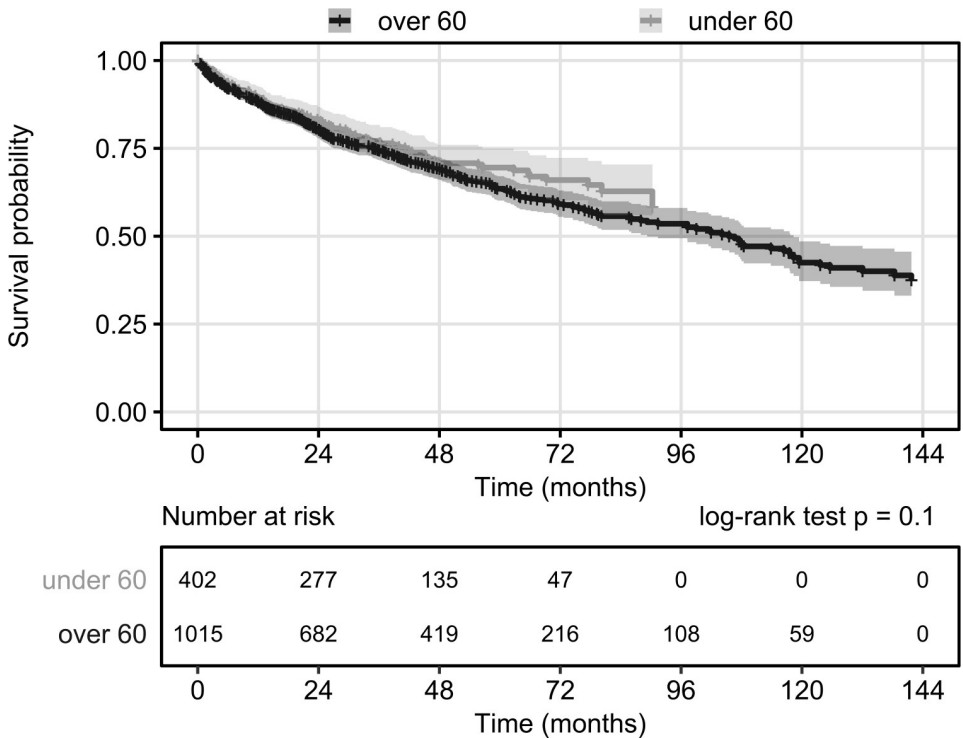

**Fig 8. Overall survival of CRC patients amongst the UC patients from CRC diagnosis by age.** Start: CRC diagnosis date. Event: death. Censoring: end of data availability (end of 2016). Shaded areas denote 95% confidence bands.

the onset of ulcerative colitis (30–39 years and over 50 years) was found in present study, which is similar to the findings of the Inflammatory Bowel South-Eastern Norway (IBSEN) Study Group [16]. In our study, the median age at diagnosis was higher than in previous studies, which is probably coming from the second peak of onset.

It is still questionable whether UC-patients are at higher risk of dying in contrast with the general population or not. Overall and cause-specific mortality was assessed by a meta-analysis of population-based inception cohort studies. They found that the overall mortality of UC patients was similar to the general population (the overall standardized mortality ratio of 1.1 (95% CI: 0.9–1.2, p = 0.42)); however, the cause-of-death distribution seemed to be different, with a higher risk of gastrointestinal diseases [10]. In our study only the overall mortality was assessed.

In contrast to the above-mentioned meta-analysis, our study revealed a significant difference (HR = 1.65) between the survival of patients with UC and that of the general population.

The most common malignancies of the Hungarian UC population were CRC, malignant neoplasm of skin, and malignant neoplasm of the prostate. These findings are in accordance with the results of a Danish population-based cohort study between 1962 and 1987. They found an increased risk for colorectal cancer and among men, for melanoma, but no increased risk for other cancers could be detected [17].

In our study 8.5% of the incident UC population were diagnosed with colorectal carcinoma between 2010–2016. The median age of the diagnosis of colorectal cancer was 65.8 years, which is higher than the average age previously published from Hungary, but similar to the general population [7]. Comparing the survival of younger and older populations, no significant difference was found between patients who were diagnosed with CRC at an age below or

above 60. A study from 2006 found the incidence of CRC 2.5%, 7.6% and 10.8% after 20, 30 and 40 years of disease duration of UC, respectively [18]. A Swedish study of colonoscopic surveillance for UC found lower risk of CRC development, 2.0%, 3.0% and 9.4% at 20, 30 and 40 years. They found a threefold increased risk of CRC compared to the general population [19].

Our study has some strengths and limitations that should be mentioned. A nationwide claims and insurance database was used in the study which is based on the sole insurance fund in Hungary with close to full population coverage. A major limitation is the retrospective nature of the study, where the primary aim of the data collection was not the clinical evaluation of patients, but rather to serve financial and reimbursement purposes. Therefore, no data was available on clinical outcomes, such as laboratory values, disease severity indices, access to healthcare or patient reported outcomes. Dosing information on all pharmaceutical products was limited. Only all-cause deaths data could be analyzed, because cause of death data (cancer or disease-specific) were not available for all deaths and are highly inconsistent even when they were available. Therefore, cause of death was decided not to be analyzed in this study. Another major limitation of our study was the limited amount of information on the matched general population–only the date of death could be obtained in addition to the age (date of birth) and gender which the matching was based on.

## Conclusion

In conclusion, our nationwide, population-based study was the first to estimate the different malignancies and mortality data among Hungarian ulcerative colitis patients, and also updated previously available data of prevalence and incidence rates. Although the mortality trend of the Hungarian UC population is in concordance with the trend observable in the general population of Hungary, our results revealed a significantly worse survival of UC patients suffering from CRC than that of the general population. These findings emphasize the importance of colorectal cancer surveillance program in the management of UC.

## Supporting information

**S1 File. Minimal dataset derived from the database.** Sheet "Prevalence, Incidence"–Prevalent and incident patient numbers. Sheet "Demographics"–Patient counts in age groups, median, mean and standard deviation of age. Sheet "Deaths"–Raw death counts and median age at death. Sheet "Malignancies distribution"–Patient counts with malignancy diagnoses based on 3-digit ICD-10 code. Sheet "CRC epidemiology"–Patient counts with new CRC diagnoses, median age at CRC diagnosis and death, total death counts. Sheet "Survival1"–Survival curve data, OS, UC patients and controls. Sheet "Survival2"–Survival curve data, OS, CRC-UC patients, whole group and age stratified.
(XLSX)

## Author Contributions

**Conceptualization:** Ágnes Milassin, Judit Gimesi-Országh, András Borsi, Péter L. Lakatos, Tamás Szamosi.

**Data curation:** Péter Kunovszki, Judit Gimesi-Országh, Péter Takács, Péter L. Lakatos, Tamás Molnár.

**Formal analysis:** Péter Kunovszki, Judit Gimesi-Országh.

**Funding acquisition:** Péter Takács, András Borsi.

**Methodology:** Péter Kunovszki, Judit Gimesi-Országh, Péter L. Lakatos, Tamás Molnár.

**Project administration:** Judit Gimesi-Országh.

**Supervision:** Péter Takács, András Borsi.

**Validation:** Ágnes Milassin, Péter Takács, Kata Szántó, Anita Bálint, Klaudia Farkas, Péter L. Lakatos, Tamás Molnár.

**Writing – original draft:** Péter Kunovszki, Ágnes Milassin, Judit Gimesi-Országh, Kata Szántó, Anita Bálint, Klaudia Farkas, Tamás Molnár.

**Writing – review & editing:** Péter Kunovszki, Ágnes Milassin, Judit Gimesi-Országh, Kata Szántó, Anita Bálint, Klaudia Farkas, András Borsi, Péter L. Lakatos, Tamás Molnár.

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
