## [Decision Letter · Decision Letter 0]

31 Dec 2019

PONE-D-19-32655

Epidemiology, mortality and prevalence of colorectal cancer in ulcerative colitis patients between 2010-2016 in Hungary – a population-based study

PLOS ONE

Dear Mr. Kunovszki,

Thank you for submitting your manuscript to PLOS ONE. After careful consideration, we feel that it has merit but does not fully meet PLOS ONE’s publication criteria as it currently stands. Therefore, we invite you to submit a revised version of the manuscript that addresses the points raised during the review process.

We would appreciate receiving your revised manuscript by Feb 14 2020 11:59PM. To enhance the reproducibility of your results, we recommend that if applicable you deposit your laboratory protocols in protocols.io, where a protocol can be assigned its own identifier (DOI) such that it can be cited independently in the future. For instructions see: http://journals.plos.org/plosone/s/submission-guidelines#loc-laboratory-protocols

We look forward to receiving your revised manuscript.

Kind regards,

Valérie Pittet, PhD

Academic Editor

PLOS ONE

2. We noticed you have some minor occurrence(s) of overlapping text with the following previous publication(s), which needs to be addressed:

https://insights.ovid.com/pubmed?pmid=17156150

https://doi.org/10.1016/j.crohns.2014.06.004

https://www.ncbi.nlm.nih.gov/pubmed/10231311

In your revision ensure you cite all your sources (including your own works), and quote or rephrase any duplicated text outside the Methods section. Further consideration is dependent on these concerns being addressed.

3. In the ethics statement in the manuscript and in the online submission form, please provide additional information about the patient records/samples used in your retrospective study. Specifically, please ensure that you have discussed whether all data/samples were fully anonymized before you accessed them and/or whether the IRB or ethics committee waived the requirement for informed consent. If patients provided informed written consent to have data/samples from their medical records used in research, please include this information.

"I have read the journal's policy and the authors of this manuscript have the following competing interests:

PK, JGO, PT, AB are employees / consultants of Janssen.

TM received speaker's honoraria from MSD, AbbVie, Egis, Goodwill Pharma, Takeda, Pfizer and Teva."

Reviewers' comments:

Reviewer's Responses to Questions

**Comments to the Author**

1. Is the manuscript technically sound, and do the data support the conclusions?

Reviewer #1: Yes

Reviewer #2: Partly

2. Has the statistical analysis been performed appropriately and rigorously? 

Reviewer #1: Yes

Reviewer #2: No

3. Have the authors made all data underlying the findings in their manuscript fully available?

Reviewer #1: Yes

Reviewer #2: No

4. Is the manuscript presented in an intelligible fashion and written in standard English?

Reviewer #1: Yes

Reviewer #2: No

5. Review Comments to the Author

Reviewer #1: Thank you for the opportunity to review the manuscript "Epidemiology, mortality and prevalence of colorectal cancer in ulcerative colitis patients between 2010-2016 in Hungary – a population-based study" Please find my comments below.

1. As many may be unfamiliar with your healthcare database it would be helpful to have either a better description or a reference to a description of the database in the manuscript and some highlights as it pertains to this manuscript.

2. Endpoints need to be described better. Primary endpoints are endpoints that your study is powered to detect. Secondary endpoints are those which the study is not powered to detect. Just stating that you had 3 endpoints is not adequate.

3. I would advocate removing figure 2 altogether since you state that you cannot adequately determine the incidence. You do a nice job of explaining why, but that essentially renders most of your data in that table questionable at best. I think keeping he descriptive language is good enough for this.

4. On Figure 3, the legend at the top has the male label over the female plot and vv. I would recommend switching that order to make it easier to understand.

5. Figure 5 is too difficult to interpret. Please either include the names on the cancers on the Y axis or highlight and name some key cancers within the figure.

6. Are deaths cancer-specific, disease-specific, or all cause? It seems like they are all cause but I think this needs to be highlighted better. This is also a big weakness of this manuscript because it is hard to know if these cancer patients had more severe disease, or accessed healthcare less and had other comorbidities. This should be discussed more in depth in the discussion.

7. Figure 8 is labeled poorly. It seems like it is CRC patients with UC, but from the labeling it appears to be all CRC patients in the country. This labeling should be improved upon.

8. It would be important to know whether the change in mortality from CRC in UC was unique to UC or whether this was similar as to the general population. This data should be collectable using your database and I would recommend including this as well.

Reviewer #2: This is a population-based national study on UC and associated CRC mortality, among others. Administrative data is used. The authors find an increased risk of CRC-mortality in UC patients.

Abstract

1. There is no clear description of the analytical methods used, including type of analysis, matching procedure.

2. The aim in the methods section should be more appropriately located in the background section.

General

1. The manuscript needs to be edited by a thorough English speaker.

Introduction

1. The introduction is winding and overly long and could be shortened considerably. What does this study add, why was it done? What gap of knowledge exists and how could this study help`?

Methods

2. The aim should be placed in the introduction section, along with a clear hypothesis.

3. The English is not up to par in the "Data Collection" section, making it difficult to follow. PLease revise.

4. The described algorithm for identifying UC patients seems reasonable, though validity is not mentioned. Has this algorithm been tested against e.g. chart data, and with what results?

5. Though I assume this to be the case, it should be mentioned that also the matched controls were also assessed for CRC using the described method for the UC patients (the cases).

6. In the statistics section, there is no mention of missing data and how this was handled. Also, was the Cox model tested for assumptions?

Results

7. The authors state that the decrease seen in UC incidence is a function of their diagnosis capture. This should be stated in the discussion, and how sure are the authors of this fact? MOreover, please analyse this statistically also (p value for difference/trend).

8. It would be interesting to see the incidence of CRC in the matched population also.

9. State p value for difference when comparing median survival in CRC UC versus non-CRC UC.

10. Survival curves should have Life tables and log rank tests, perhaps also confidence intervals.

Discussion

11. There is a paucity of other populationbased studies (apart from Hungarian ones) in the discussion. E.g. Rutegard 2016 Scand J Surg found 9% CRC incidence after 38 years of screening colonoscopy in a defined UC cohort and should be cited, along with the St Marks study (Rutter 2006 Gastroenterology). Is there any screening programmes for UC in Hungary?

6. PLOS authors have the option to publish the peer review history of their article (what does this mean?). If published, this will include your full peer review and any attached files.

Reviewer #1: No

Reviewer #2: No

---

## [Author Response · Author response to Decision Letter 0]

25 Feb 2020

We have updated the style of the manuscript, including numbering lines and reformatting figure labels.

2. We noticed you have some minor occurrence(s) of overlapping text with publication(s).

We rephrased the above-mentioned parts of the manuscript. However, we made an effort to find the original article from

Saro Gismera C. et al. (1999) (https://www.ncbi.nlm.nih.gov/pubmed/10231311),

but since it was written in Spanish, and we don’t speak Spanish, we were unable to find and rephrase the overlapping parts with our manuscript.

3. In the ethics statement in the manuscript and in the online submission form, please provide additional information about the patient records/samples used in your retrospective study. Specifically, please ensure that you have discussed whether all data/samples were fully anonymized before you accessed them and/or whether the IRB or ethics committee waived the requirement for informed consent. If patients provided informed written consent to have data/samples from their medical records used in research, please include this information.

We have added the necessary information to both the manuscript and submission form.

The research team had access to anonymized data, the anonymization was performed by the data holder. The re-use of public data (which includes the data used in the study) is guaranteed by law in Hungary (based on Act 63/2012 on the re-use of public data).

The research group had access to the data indirectly, through NHIF according to internal data privacy regulations of NHIF and Regulation (EU) 2016/679 General Data Protection Regulation (GDPR). Due to this, and to the retrospective nature of the study there was no need for patient level consent to the analysis.

4. Competing interests.

We would like to modify the competing interests statement to the following due to declarations that I had received after the initial submission was done:

I have read the journal’s policy and the authors of this manuscript have the following competing interests:

PK, JGO, PT, ABo are employees / consultants of Janssen

PLL has been a speaker and/or advisory board member for AbbVie, Arena Pharmaceuticals, Celltrion, Falk Pharma GmbH, Ferring, Genetech, Janssen, Merck, Pharmacosmos, Pfizer, Roche, Shire and Takeda and has received unrestricted research grants from AbbVie, MSD and Pfizer.

TSz has served as advisory board member for AbbVie, EGIS, Pfizer and Takeda, received speaker’s honoraria from Abbvie, Takeda and Ferring and served as part time medical advisor for Hungarian National Health Insurance Fund.

TM received speaker’s honoraria from MSD, AbbVie, Egis, Goodwill Pharma, Takeda, Pfizer and Teva.

KF received speaker’s honoraria from AbbVie, Janssen and Ferring.

ABá: received speaker’s honoraria from Janssen and Ferring.

KSz, ÁM have no conflicts of interest to declare.

This does not alter our adherence to PLOS ONE policies on sharing data and materials.

5. Data access

The dataset that was used is held by the National Health Insurance Fund (NHIF) of Hungary (http://www.neak.gov.hu, e-mail: neak@neak.gov.hu).

Access to the individual-level data is available after filing a formal data access request to adatkeres@neak.gov.hu. Requestors need to accept the terms and conditions of the data request and may need to pay the corresponding data access fee.

The terms of the contract for data access does not allow the reporting of any data of a single individual or results which comes from aggregating the data of less than 10 individuals. Therefore, a de-identified dataset could not be provided. Taking these requirements into consideration, the results can be published.

A supplementary dataset was created which contains the patient counts derived from the original data.

Reviewers’ comments:

Reviewer #1:

1. As many may be unfamiliar with your healthcare database it would be helpful to have either a better description or a reference to a description of the database in the manuscript and some highlights as it pertains to this manuscript.

We revised the manuscript and added some information in the Data collection section about the database used.

2. Endpoints need to be described better. Primary endpoints are endpoints that your study is powered to detect. Secondary endpoints are those which the study is not powered to detect. Just stating that you had 3 endpoints is not adequate.

We absolutely agree with the Reviewer that the phrasing „endpoint” is not the most appropriate word we used, as our study is purely descriptive in nature. We decided not to use this terminology at all and rephrased the above-mentioned part of the manuscript and moved these parts to the Methods section.

3. I would advocate removing figure 2 altogether since you state that you cannot adequately determine the incidence. You do a nice job of explaining why, but that essentially renders most of your data in that table questionable at best. I think keeping he descriptive language is good enough for this.

Based on the Reviewer’s suggestion, we revised the above-mentioned part of the Manuscript. Figure 2 was removed, and the paragraph has been shortened as well.

4. On Figure 3, the legend at the top has the male label over the female plot and vv. I would recommend switching that order to make it easier to understand.

We made the suggested switch in the labelling.

5. Figure 5 is too difficult to interpret. Please either include the names on the cancers on the Y axis or highlight and name some key cancers within the figure.

Using some abbreviations, it was possible to include the name of cancers on the Y axis. The caption of the figure was revised accordingly.

6. Are deaths cancer-specific, disease-specific, or all cause? It seems like they are all cause but I think this needs to be highlighted better. This is also a big weakness of this manuscript because it is hard to know if these cancer patients had more severe disease, or accessed healthcare less and had other comorbidities. This should be discussed more in depth in the discussion.

We revised and completed the discussion and the limitations parts of the manuscript.

The primary aim of the data collection was not the clinical evaluation of patients, but rather to serve financial and reimbursement purposes. Therefore, no data were available on clinical outcomes, such as laboratory values, disease severity indices, access to healthcare or patient reported outcomes.

7. Figure 8 is labeled poorly. It seems like it is CRC patients with UC, but from the labeling it appears to be all CRC patients in the country. This labeling should be improved upon.

We improved the labeling accordingly.

8. It would be important to know whether the change in mortality from CRC in UC was unique to UC or whether this was similar as to the general population. This data should be collectable using your database and I would recommend including this as well.

Unfortunately, no information other than the date of birth, date of death and gender could be obtained for the general population in this study; therefore, no further analyses regarding these controls were possible.

We have clarified this circumstance in the Data collection and Discussion parts of the manuscript.

Reviewer #2:

1. Abstract – There is no clear description of the analytical methods used, including type of analysis, matching procedure.

The study is descriptive in nature. We added this fact to the abstract. We also added some description of the matching procedure to the abstract.

2. Abstract – The aim in the methods section should be more appropriately located in the background section.

We made the change proposed, the aims were moved to the Introduction section.

The manuscript needs to be edited by a thorough English speaker.

We revised the language used and the manuscript was edited by a thorough English speaker. We hope it matches the journal standard.

1. The introduction is winding and overly long and could be shortened considerably. What does this study add, why was it done? What gap of knowledge exists and how could this study help`?

We have shortened the Introduction section by moving some of the content to the Discussion section. The background of the study and the aims of the study are clearly indicated in the revised text.

2. The aim should be placed in the introduction section, along with a clear hypothesis.

We moved the aim section to the end of the introduction. Our study is a descriptive, retrospective, epidemiological study; therefore, no formal hypotheses were formulated.

3. The English is not up to par in the "Data Collection" section, making it difficult to follow. Please revise.

We have revised the “Data Collection” section by adding more details to make it easier to understand. Some sentences were also deleted or rephrased.

4. The described algorithm for identifying UC patients seems reasonable, though validity is not mentioned. Has this algorithm been tested against e.g. chart data, and with what results?

The algorithm is originated from a previous Hungarian study by Kurti et al. (doi: 10.1016/j.dld.2016.07.012). We lacked the necessary data to validate this algorithm ourselves.

5. Though I assume this to be the case, it should be mentioned that also the matched controls were also assessed for CRC using the described method for the UC patients (the cases).

Unfortunately, no other data (comorbidities, etc.) except the date of birth, gender and the date of death could be obtained for the general population in this study; therefore, further analysis of the matched controls couldn’t be performed.

We have clarified this circumstance in the Data collection and Discussion parts of the manuscript.

6. In the statistics section, there is no mention of missing data and how this was handled. Also, was the Cox model tested for assumptions?

The Statistical analysis section was improved by inserting the following information:

Due to the claims nature of the data, missing data are undiscoverable in most cases. If a certain intervention or diagnosis was not recorded, there is no chance that it could be imputed in any way. Therefore, no handling of missing data was performed.

The proportional hazards assumption for the Cox model was tested using plots of Schoenfeld residuals.

7. The authors state that the decrease seen in UC incidence is a function of their diagnosis capture. This should be stated in the discussion, and how sure are the authors of this fact? Moreover, please analyze this statistically also (p value for difference/trend).

Due to the unreliability of the results obtained, we have decided to reduce the information presented. Fig 2. was removed altogether and only a point prevalence for the year 2015 was presented. As this is explained in the results section, we did not see the need to repeat it in the discussion section.

8. It would be interesting to see the incidence of CRC in the matched population also.

Unfortunately, no other data (comorbidities, etc.) except the date of birth, gender and the date of death could be obtained for the general population in this study; therefore, further analysis of the matched controls couldn’t be performed.

We have clarified this circumstance in the Data collection and Discussion parts of the manuscript.

9. State p value for difference when comparing median survival in CRC UC versus non-CRC UC.

Due to the obvious differences in the survival of CRC patients and all patients (5-year survival rates of 65% vs 86%), this formal analysis was not performed.

10. Survival curves should have Life tables and log rank tests, perhaps also confidence intervals.

Life tables were added to the corresponding figures. The p-value for log-rank tests was also included to figures with comparisons. Shading was used to display the confidence intervals for the survival curves.

11. There is a paucity of other population-based studies (apart from Hungarian ones) in the discussion. E.g. Rutegard 2016 Scand J Surg found 9% CRC incidence after 38 years of screening colonoscopy in a defined UC cohort and should be cited, along with the St Marks study (Rutter 2006 Gastroenterology). Is there any screening programmes for UC in Hungary?

We revised the discussion section and added the mentioned population-based studies.

The CRC-screening program has been started in 2019, before that, some pilot studies were conducted. No special UC-CRC screening program is available yet in Hungary, but we follow the ECCO-guidelines on outpatient follow-up visits.

---

## [Decision Letter · Decision Letter 1]

16 Mar 2020

PONE-D-19-32655R1

Epidemiology, mortality and prevalence of colorectal cancer in ulcerative colitis patients between 2010-2016 in Hungary – a population-based study

PLOS ONE

Dear Mr. Kunovszki,

Thank you for submitting your manuscript to PLOS ONE. After careful consideration, we feel that it has merit but does not fully meet PLOS ONE’s publication criteria as it currently stands. Therefore, we invite you to submit a revised version of the manuscript that addresses the points raised during the review process.

We would appreciate receiving your revised manuscript by Apr 30 2020 11:59PM. To enhance the reproducibility of your results, we recommend that if applicable you deposit your laboratory protocols in protocols.io, where a protocol can be assigned its own identifier (DOI) such that it can be cited independently in the future. For instructions see: http://journals.plos.org/plosone/s/submission-guidelines#loc-laboratory-protocols

We look forward to receiving your revised manuscript.

Kind regards,

Valérie Pittet, PhD

Academic Editor

PLOS ONE

Reviewers' comments:

Reviewer's Responses to Questions

**Comments to the Author**

1. If the authors have adequately addressed your comments raised in a previous round of review and you feel that this manuscript is now acceptable for publication, you may indicate that here to bypass the “Comments to the Author” section, enter your conflict of interest statement in the “Confidential to Editor” section, and submit your "Accept" recommendation.

Reviewer #1: All comments have been addressed

Reviewer #2: All comments have been addressed

2. Is the manuscript technically sound, and do the data support the conclusions?

Reviewer #1: Yes

Reviewer #2: Yes

3. Has the statistical analysis been performed appropriately and rigorously? 

Reviewer #1: Yes

Reviewer #2: Yes

4. Have the authors made all data underlying the findings in their manuscript fully available?

Reviewer #1: No

Reviewer #2: Yes

5. Is the manuscript presented in an intelligible fashion and written in standard English?

Reviewer #1: Yes

Reviewer #2: Yes

6. Review Comments to the Author

Reviewer #1: Thank you for addressing our concerns. The introduction is still quite difficult to read and I would recommend having a native English speaker rewrite it prior to acceptance.

Reviewer #2: The authors have responded adequately to most questions and I can only recommend publication. Please note that Figure 6 seems to be inverted, i.e. have cases and controls been mixed up? Seems strange that the colitis should be both bigger and more prone to survive, given the increased HR for this group stated in the manuscript.

7. PLOS authors have the option to publish the peer review history of their article (what does this mean?). If published, this will include your full peer review and any attached files.

Reviewer #1: No

Reviewer #2: No

---

## [Author Response · Author response to Decision Letter 1]

29 Apr 2020

Reviewer #1:

The Reviewer indicated that the authors did not make all data underlying the findings in the manuscript fully available.

Please note that due to legal and ethical restrictions, the patient level data could not be made available. Please also note that a supplementary dataset containing aggregated data based on the original patient-level data was created and made available with the submission.

Please refer to the data availability statement (quoted below) for further information.

The dataset that was used is held by the National Health Insurance Fund (NHIF) of Hungary (http://www.neak.gov.hu, e-mail: neak@neak.gov.hu).

Access to the individual-level data is available after filing a formal data access request to adatkeres@neak.gov.hu. Requestors need to accept the terms and conditions of the data request and may need to pay the corresponding data access fee.

The terms of the contract for data access does not allow the reporting of any data of a single individual or results which comes from aggregating the data of less than 10 individuals. Therefore, a de-identified dataset could not be provided. Taking these requirements into consideration, the results can be published.

A supplementary dataset was created which contains the patient counts derived from the original data.

The introduction is still quite difficult to read and I would recommend having a native English speaker rewrite it prior to acceptance.

The whole manuscript was checked for stylistic and grammatical errors and some minor changes were made throughout.

Reviewer #2:

Please note that Figure 6 seems to be inverted, i.e. have cases and controls been mixed up? Seems strange that the colitis should be both bigger and more prone to survive, given the increased HR for this group stated in the manuscript.

Thank you for catching our mistake. Some kind of technical error occurred while creating the graph. We have addressed the issue and the graph is resubmitted.

---

## [Editor Report · Decision Letter 2]

1 May 2020

Epidemiology, mortality and prevalence of colorectal cancer in ulcerative colitis patients between 2010-2016 in Hungary – a population-based study

PONE-D-19-32655R2

Dear Dr. Kunovszki,

We are pleased to inform you that your manuscript has been judged scientifically suitable for publication and will be formally accepted for publication once it complies with all outstanding technical requirements.

With kind regards,

Valérie Pittet, PhD

Academic Editor

PLOS ONE
---

## [Editor Report · Acceptance letter]

5 May 2020

PONE-D-19-32655R2 

Epidemiology, mortality and prevalence of colorectal cancer in ulcerative colitis patients between 2010-2016 in Hungary – a population-based study 

Dear Dr. Kunovszki:

I am pleased to inform you that your manuscript has been deemed suitable for publication in PLOS ONE. Congratulations! Your manuscript is now with our production department. 

With kind regards,

on behalf of

PD Dr. Valérie Pittet 

Academic Editor

PLOS ONE